# Translation in Digital Times: Omid Tofighian on Translating the Manus Prison Narratives

**Omid Tofighian** [1,2]

[1] School of the Arts and Media, University of New South Wales, Kensington, NSW 2052, Australia; omid_tofighian@yahoo.com

[2] School of Law, Birkbeck, University of London, London WC1E 6DP, UK

**Abstract:** On 12 February 2020, while on an international tour promoting Behrouz Boochani's *No Friend but the Mountains: Writing from Manus Prison*, the translator of the book, Omid Tofighian, participated in a seminar at Utrecht University, organised by Australian academic, Anna Poletti (associate professor of English language and culture, Utrecht University). Poletti is also co-editor of the journal *Biography: an interdisciplinary quarterly*, which published a special issue on *No Friend but the Mountains* in 2020 (Vol. 43, No. 4). The seminar involved Poletti, Tofighian and translation scholar, Onno Kosters (assistant professor of English literature and translation studies, Utrecht University) in conversation. Iranian–Dutch filmmaker, Arash Kamali Sarvestani, co-director with Boochani of the film *Chauka, Please Tell Us the Time* (2017), was in attendance, as well as the Dutch publisher, Jurgen Maas (Uitgeverij Jurgen Maas, Dutch translation based on the English translation). The event was titled 'No Friend but the Mountains: Translation in Digital Times'. The following dialogue, 'Translation in Digital Times: Omid Tofighian on Translating the Manus Prison Narratives', is derived from this seminar and focuses on Tofighian's translation of the book from Persian/Farsi into English. The topics covered also include the Dutch translation from Tofighian's English translation, genre and anti-genre, horrific surrealism, Kurdish elements and influences, the Kurdish translation (from Tofighian's English translation), publication of the Persian/Farsi original, translation as activism, process and technology.

**Keywords:** refugees; exile; Manus Island; Australia; Behrouz Boochani; translation; literature

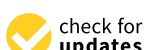



*No Friend but the Mountains: Writing from Manus Prison* (Boochani 2018) was written by Behrouz Boochani during the first five years of his incarceration in the Australian-run offshore immigration detention centre on Manus Island, Papua New Guinea—a former colony of Australia. It was written completely on WhatsApp via hundreds of text messages and sent to friends and colleagues in Australia for translation, editing and publication (Persian/Farsi to English). Moones Mansoubi collated most of the messages into individual chapters and created PDFs for Omid Tofighian to edit and translate; Boochani continued to send text messages to Tofighian to insert/change the text while they worked on it as a cocreation; the final chapter arrived directly to Tofighian via one text. The final chapter (Chapter 12: 'In Twilight/The Colours of War') was completed during the 23-day siege and forced removal of refugees to new prison camps in October–November 2017 (see Boochani Forthcoming, chp. 5 for reports and critical commentary about the siege). *No Friend but the Mountains* was awarded the 2019 Victorian Prize for Literature among many other prestigious awards, and Boochani (through video link) was invited to speak at many Australian and international festivals, book launches, seminars, campaign events and conferences together with his translator and collaborator, Tofighian (mostly in person). Since the release of the book, a series of events have focused on translation (for a comprehensive account of the history and context associated with Boochani's plight and his collaboration with Tofighian see the introduction in Tofighian's article published in this issue titled 'On Representing Extreme Experiences in Writing and Translation: Omid

Tofighian on Translating the Manus Prison Narratives' (Tofighian 2022), https://www.mdpi.com/2076-0787/11/6/141 accessed on 10 November 2022).

On 12 February 2020, while on an international tour promoting the book, Tofighian participated in a seminar at Utrecht University, organised by Australian academic, Anna Poletti (associate professor of English language and culture, Utrecht University). Poletti is also co-editor of the journal *Biography: an interdisciplinary quarterly*, which published a special issue on *No Friend but the Mountains* in 2020 (vol. 43, No. 4; Poletti 2020). The seminar involved Poletti, Tofighian, and translation scholar, Onno Kosters (assistant professor of English literature and translation studies, Utrecht University) in conversation. Iranian–Dutch filmmaker, Arash Kamali Sarvestani, codirector with Boochani of the film *Chauka, Please Tell Us the Time* (Boochani and Sarvestani 2017), was in attendance, as well as the Dutch publisher, Jurgen Maas (Uitgeverij Jurgen Maas, Dutch translation based on the English translation). The event was titled: 'No Friend but the Mountains: Translation in Digital Times'.

The following dialogue, 'Translation in Digital Times: Omid Tofighian on Translating the Manus Prison Narratives', is derived from this seminar and focuses on Tofighian's translation of the book from Persian/Farsi into English. The topics covered also include the Dutch translation from Tofighian's English translation, genre and antigenre, horrific surrealism, Kurdish elements and influences, the Kurdish translation (from Tofighian's English translation), publication of the Persian/Farsi original, and translation as activism, process and technology (see Supplementary Materials for a reading list of articles by Tofighian and Boochani about related themes and issues).

**Utrecht University, the Netherlands 2020**

> **Onno:** I read the English and the Dutch translation.[1] I am in touch with the translator, Irwan Droog. You say the last chapter was written in epic form—chapter twelve. It has long stretches of text that really look like poetry and definitely also read like poetry. But there are also long stretches of prose and that is one of the things that struck me about the entire book; that is, it is so diverse in the genres that it includes. It has activist prose, observation, poetry and more. And I was wondering from a translator's point of view whether those various genres offered any particular difficulties, particularly the way in which you deal with translation issues that you come across. Was there a big difference in how you tackled the various problems in the various genres?

> **Omid:** I think this is an excellent question and especially something that really dives right into the peculiarities of translation, or the difficulties of translation. So I appreciate it. Could I say something about genre first and then I will move on to talk about the translation style, the decisions made and the difficulties in translating?

> So first of all, regarding genre, the last chapter is written in epic form but has other genres mixed into it. But that is a characteristic of the whole book. The whole book has a mixture of political commentary, aspects of Behrouz's journalism, it has philosophical ruminations, psychoanalytic examination, poetry, it has realistic accounts of what was happening. But mixed into all of this—and somehow also structuring these other genres—is myth, epic and folklore. You could read this book as a modern epic. You could read it also as a contemporary or an updated form of Kurdish folklore. So much of this book is based on the stories Behrouz grew up with, what he learned from his mother, first of all, from his village and also from Kurdish history, literature and resistance.

> **Onno:** You wrote about that in your supplementary essays in *No Friend but the Mountains*. Yes.

> **Omid:** And working with Behrouz, I had to do a lot of research and came across a lot of really important Kurdish writers. I noticed how deeply influential their

work has been on Behrouz's thinking and writing. So regarding genre, I call it an anti-genre. So much about this book resists the principles, assumptions and norms associated with genres. It basically challenges us to rethink exactly how or why we use categorisation the way we do in the first place. So I call it an anti-genre. And I was trying to think of a style or maybe a framework or hermeneutical schema with which to position this book, because I did not want to use genres to describe it. So I coined the term 'horrific surrealism' (Tofighian 2018b, 2021) for two reasons; first of all, the mixture of horror realism and psychological horror with surrealism is characteristic of not only his own positionality and his personhood but also the circumstances, the Australian political situation, even our relationship with each other. But also, it says so much about the contemporary Kurdish situation or the Kurdish plight, the Kurdish struggle. And you find the mixture of horror and surrealism in other Kurdish writers as well like Sherko Bekas, who Behrouz is deeply influenced by; Bekas uses these features quite a lot. Also, horror and surrealism are fundamental in the writings of Sherzad Hassan, who Behrouz also reads. Kurdish filmmakers like Bahman Ghobadi, who is a Kurd from Iran—you also find them in his films (Tofighian 2018a). So this mixture of horror and surrealism I found really interesting, not as a genre but maybe as an attitude, a vision or a schema through which to think about the book.

**Onno:** That is a very powerful mode, of course, a very powerful way of seeing it. I am thinking of the film, *Parasite*, that many of you would have seen, which is a satire and also has kind of a horrific surrealism aspect, as well. So it is a very powerful way of showing what is actually going on and bringing across the message.

**Omid:** Definitely, another thing that I like about the reference to surrealism and horror is that it links the book to other anti-colonial forms of resistance that use art and literature, particularly in the Caribbean, other examples of the African diaspora, also within Africa. So consider the Négritude movement and then later with Afro–Surrealism, for instance. And so there you find very similar themes. Behrouz is not familiar with those works, but his anti-colonial perspective, his way of being, and also his Indigenous Kurdish identity, all these elements feed into each other in similar ways.

**Onno:** So it is something that Salman Rushdie, of course, would also mix in all his works, the idea of the surreal, the real and the mythical.

**Omid:** Although I think that Salman Rushdie and many others say, for instance, Toni Morrison and I think here in the Netherlands with Kader Abdalah and Abdelkader Benali, use more magical realism. This was not a distinction I was making in the beginning when I started translating the book, I thought Behrouz was only employing magic realism or magical realism (even though there are many similarities and overlaps with surrealism). But then I had a conversation with a colleague in Cairo—I was working in Cairo for a couple of years and moved there when I was half-way through the translation—who is an expert on magical realism. And when I was telling her about this book, she said, that is interesting, it is great and there are some overlaps, but it is not magical realism. And so we were involved in a really long and interesting discussion about this. And I realised that unlike magical realist writers that assume the world is already wonderful, fantastic, magical, *No Friend but the Mountains* talks about the subconscious mind creating these sorts of images, scenes and centres Behrouz's interpretations. Also, the fragmentation and the dream visions put this work maybe on the cusp of magical realism, but more into the area of surrealism.

**Anna:** So just to pick up on the second part of Onno's question, was horrific surrealism the kind of overall schema that you used to negotiate the challenge of translating all these different genres? Is this part of the consistent approach and

the way to both retain what is special about philosophical reflection and poetry? That is, you retain both the specific contributions to the work, but you keep them consistent through horrific surrealism?

**Omid:** Yes, particularly because other aspects of surrealism are fragmentation, disjunction, disjointedness, what is shattered.

**Anna:** Can I pick up on something that you said that links into one of my questions? I thought it was really interesting when you were listing all the things that horrific serialism captured both about the text itself and about the process of its production, and you said *including our relationship*. So I am interested in that. What is the horrifically surreal nature of the relationship between the translator and the writer, particularly under the conditions that you outlined for us? So could you talk a little bit about how your practice of translation is also an intellectual collaboration. How is your relationship with Behrouz an example of horrific surrealism? And then talk to us a little bit about your translator's note and translator's reflection in which you talked quite a bit about intellectual collaboration, political solidarity and aesthetic interpretation as being key to your approach to translation. Could you tell us whether that philosophy of translation emerged out of the practices itself, as you were doing it? Was that how you worked out what your principles of translation were, or whether that was already your kind of philosophy of translation, and you were using those principles to inform your decision making?

**Omid:** I will start with that and then go back to the specific surreal aspect of our collaborative work.

We did not have time to really think deeply about everything we were doing and planning. Everything was so urgent. Everything was so pressured. There was just too much going on at the same time. I also had a life, you know, other things that I was doing. And it basically meant that for the last four and a half years I have not really had a break. I had to deal with my own responsibilities, my own commitments, and whenever I had time—and if I did not have time, I made time—I worked on Behrouz's writing and also the work of other people on Manus Island and in other Australian offshore and onshore detention centres. So I am working with others there as well, writers and other creatives and intellectuals. Everything was basically born out of lived experience. Everything emerged out of our interactions, out of our resistance. I was learning from Behrouz, I was sharing things with him, and vice versa. We were discussing, we were debating. We could not plan one particular strategy for activism because the minute we actually came up with something that we thought would work, would challenge the system . . . the system changed. And this kind of absurdity, this irony, planning without any certainty, without the hope for any specific outcome, is also part of the surreal nature of the whole situation. I do not have a clear answer for you because nothing was planned, and so much was just spontaneous, ad hoc. But what we did do, and what I can say for certain, is we identified all the gaps and weaknesses in the system, and also the gaps and weaknesses in activism. So we knew what we should not be doing, what does not work, what ends up being exploitative, what ends up being pointless.

**Onno:** Could you give us an example?

**Omid:** One thing that I realized from the early 2000s when I was an undergraduate and engaged in very early and basic forms of activism was that I was not comfortable working with any particular organisations or groups. I was let down a number of times. I was kind of disappointed with some of the behaviours and responses. There is also a kind of Orientalism in some circles that I felt uncomfortable with. I mean, in activist circles. And I realised that I could do so much more working with individuals, smaller groups and communities, people

who I felt were already engaged in activism, doing some of the most important forms of activism, but were never recognised as activists because . . . they are just part of a cultural community . . . they are just doing what they do because they need to be 'there'. And in many ways, for them it was a form of survival and pride. Those kinds of community outreach projects, those kinds of commitments and the dedication to their community was basically something that many people did because they did not have a choice. It was something they had to do in order to make sense of the world and to thrive. So those early experiences characterised my own activism and I left Australia soon after that time and lived in different countries—in the Emirates, in Belgium and the Netherlands for my PhD—and I learnt a lot about activism when travelling around. There are some great things happening in activist circles in the Middle East and Europe. So I returned to Australia and started working again in this space—that was at the end of 2010. I think it was about five years after that I met Behrouz and realised that something different needs to happen in this situation. In direct response to your question, one of the big gaps that I referred to is the complicity of the human rights organisations connected with the detention industry; when we are talking about immigration detention in Australia, we are talking about a business. This is about racial capitalism. This is about neoliberal policies. This is about people making money. This is about the system, the border-industrial complex. This is something we cannot really talk about a lot now, but we will in the future. Behrouz and I have talked a lot about the role of human rights organisations in this industry. Maybe I will leave it there because it is so complicated, and Behrouz and I are very strategic about how and when to address this. We have a long-term plan, he had to work with these organisations in some way in order to plan his escape to Aotearoa New Zealand. So there will be a right time to talk about this and not in a kind of 'call-out way', not in a toxic kind of way or making an accusation, but in a really constructive way to find out, basically, where we need to go from here, and ask how we stop this. Because this is not just about people subject to border violence. This is also about citizens; it also affects us (Tofighian 2020).

And this leads us to your question about our relationship and the surreal nature of it. So much about my relationship with Behrouz is about not only understanding his circumstances, the kind of system that has been designed and implemented in the prison island, but also about me as a citizen. Where do I stand in this? How am I complicit? Maybe in order to talk about horrific surrealism, we need to talk about Behrouz's positionality, his own identity, the fragmented, disjointed, shattered, absurd nature of what he has gone through and how that speaks to and relates to the political situation in Australia, the absurdity of the whole system, the fluidity, all the ironies associated with a liberal democracy doing this sort of thing, conducting systematic torture. That also corresponds to our relationship. So there is this kind of circular reinforcing going on, because I get to travel across borders. He was stuck in an island prison, in a cell, and I was travelling the world showing my Australian passport all over the place, getting invited to events and attending receptions—I remember this when Arash [Kamali Sarvestani] visited Sydney from the Netherlands for the premier of *Chauka, Please Tell Us the Time* (2017, dir. Kamali Sarvestani and Boochani). I would have my drink in hand, talking to all of these people, hearing gestures of gratitude while Behrouz was still stuck in Manus. So when the phone switches off and Behrouz is off the screen, after the Skype connection or the Zoom connection ends, we all go back to our everyday lives. And then I travel the next day, or I go to another place.

**Onno:** Does that make you feel you are part of the system, as well?

**Omid:** I am part of the system. I mean, we could go into so much detail about this, because even as an academic in Australia my university superannuation until 2015 was invested in immigration detention. I was profiting from refugees while supporting them. Can you see the ironies here?

**Onno:** So are we in a way, so are we today because we are here listening to you. Oh, it is very surreal indeed.

**Omid:** Absolutely. I mean, in 2015, UniSuper said that it divested from detention, but so much still is not transparent. It is hard to actually investigate and prove it. And there are so many academic projects in universities that have relationships with the department of immigration. There are engineering and technology departments or schools working with border security initiatives to develop new technologies. So we are all complicit. This is multifaceted, multidimensional.

**Anna:** Which is horrific.

**Omid:** And I will mention that in addition to those three dimensions—Behrouz's identity, the political situation and our relationship or our interaction with each other—there are also other significant dimensions that condition and are conditioned by horrific surrealism: the structure, the content, the symbols, the tropes, and the style of the book. So everything we talked about in the beginning in response to your question Onno about anti-genre, the complexity, the confusion, the unusual juxtapositions, the incongruity, the mixture of things together, the poetics, are all part of this absurd, ironic, surreal, fragmented quality.

**Anna:** I would just like to pick up on something that you said about the process; you gave a great answer about your philosophy of translation, which is essentially I did not have one. You characterised it as a process that was defined by urgency, pressure and response to immediate challenges. This reflects digital technology, the kind of workflows and communication practices that digital mobile technology itself kind of traps us in. And there is a lot of interesting work in media studies about these very short circuits of emotion and feeling and response that mobile technology catches us in, but these are no way near as important and extreme circumstances as the ones that you were working in. It is more like, should I get shopping on the way home or not? And consider this short circuit of communication and pinging, and no one can plan anything anymore because we can all just message each other five minutes before we get there. And in terms of our interest in the role of mobile media in this process and in this product [*No Friend but the Mountains*], it is interesting to me that there is such a close alignment between the political urgency and also the way the technology itself is a technology of urgency in many ways.

**Omid:** The whole idea of horrific surrealism and different notions such as a 'shared philosophical activity' (Tofighian 2018b) or a collective agency emerged through the translation process. And all the things that you are talking about, the sound bite style communication, the quick responses or quick back and forth associated with digital technology, are relevant because Behrouz was working on WhatsApp, which is like a cell in itself. It is like a prison that he was dependent on. He ended up modifying WhatsApp, using it as a kind of Microsoft Word, he used WhatsApp like we use files on our Microsoft Office. Actually, he was using dormant WhatsApp numbers to file different pieces of writing. And because of the way he was writing on WhatsApp, I think it influenced the structure of his writing as well. I will give an example; I have a PowerPoint slide of one of the passages.

## Chapter 8
## Queuing as Torture: Manus Prison Logic /
## The Happy Cow

*A twisted, interlocking chain of hungry men /*
*Bodies mutate under the burning sun /*
*Heads in an oven fired by the sun /*
*Undergoing sickening transformations /*
*A long line of men of different heights, weights,*
*ages and colours.*

Days in the prison begin with the commotion of
long queues – long, pulverising queues .

Especially with literature, Persian/Farsi texts include very long sentences, extremely long, maybe longer and more complicated than some German texts, or maybe Dutch. The subject is at the beginning and the main verb is at the end. In between there are many clauses that go in different directions, consecutive clauses. Now, to translate that into English and maintain the same sentence structure will be messy and will lose all the rhythm, all the poetic quality, all of its profundity and style. So what I had to do was break up the long sentences into shorter sentences. This aspect and approach formed into a kind of philosophy, something that emerged out of the translation process. I decided the best way to translate these difficult parts was to split up the sentences into smaller sentences and then repeat nouns, verbs, adjectives, adverbs, phrases, etc., so that there is a clear connection between the different newly created sentences. At a certain point we realised—I think it was the third edit that we were going through—that some parts actually sound more like poems than they do prose. Behrouz only wrote prose, but a poetic style of prose. So while we were editing we thought more about my suggestion to make some of these passages that sound like poetry, actual poems in the book. The poetry parts in the book, which are in italics, were originally poetic or rhythmic prose. But as a result of the translation process, we decided to make the format poetry in English. The passage in the slide is an example of that. These images represent the surreal nature of the prison system; this passage is poetry and then converts to prose. In Farsi or Persian, that whole passage was one sentence. And you can see the repetition I employed. One sentence written in prose became this passage. Half of it, or part of it, in poetry and part of it in prose. It was all about experimentation. I am not trained as a translator, and this was my first major translation project after only some really small tasks. I had no format, or confines or formalities to abide by. I was just free. I was just doing whatever felt right. And so it was all experimental.

> **Onno:** I think it works really well. I mean, I had not realised this at all, of course. But it is because you have this kind of island hopping. That sounds really ironic, I do not mean to be ironic, but you have these poetic passages that you and Behrouz discussed, and you came up with this solution.

> **Omid:** It is part of the shared philosophical activity, we were discussing it and discussing it, and then we thought, why don't we do this and see if it works?

**Onno:** Because it really magnifies it, particularly passages like this. It really magnifies the suffering and the absurdity and surrealism, as well.

**Omid:** We thought to ourselves, it is not our purpose to give a totally realist account of what is going on. Behrouz hates the idea of his book being seen as a memoir. It is not a memoir, and the film *Chauka, Please Tell Us the Time* is not that kind of documentary. The film is not about people's lives and characters and histories. It does not go into a lot of characterisation. It is about what it feels like to live in a system that is absurd, that is designed to punish, and with technologies of systematic torture. Behrouz and Arash want you to feel that. So his purpose in the book is not to give a factual account like a journalistic piece or any kind of reportage. He is not interested in that. He says this is a work of art. He wants you to enter this world.

**Onno:** It made me think of Beckett's *Waiting for Godot*, as well.

**Omid:** I am really glad you brought that up. He was reading Beckett as he was writing. He was reading Kafka, too. He was reading Kurdish writers, Persian writers as well, and Camus. Chapter 8, 'Queuing as Torture: Manus Prison Logic/The Happy Cow' is up to thirty pages long and is about waiting in line. You know, he was reading Beckett while he was writing that particular chapter.

**Onno:** That makes sense.

**Anna:** Given this really great, illuminating example of the move from Persian/Farsi into English; Onno, I am wondering whether this is a nice moment for your question about the Dutch translation?

**Onno:** Yes, well, this is very interesting because as I understand it, Jurgen [Maas, the Dutch publisher] correct me if I am wrong here. Irwan translated this from English into Dutch, not from Persian/Farsi into Dutch.

**Jurgen:** No, because there is not a Persian/Farsi edition yet, but I just heard today that there will be a Persian/Farsi edition of the book.

**Omid:** This is an interesting story, as well. I should mention that the Kurdish translation is already out, it was actually produced by Hashem Ahmadzadeh, an academic and translator who knows English, Persian/Farsi and Kurdish [in addition to other languages] really well. He had the original raw text that I was working off and he had the English translation. However, he used the English translation to then translate into the Kurdish version, which is really unique. I do not know of any other example where this has happened.[2] About the Persian/Farsi version, Behrouz was contracted by Nashr-e Cheshmeh (or Cheshmeh Publishing House), which is one of the most reputable literary publishers in Iran.[3] Unfortunately, it will be censored.

**Jurgen:** In what way?

**Omid:** Probably for references to Kurdish struggle, Kurdish resistance.

**Jurgen:** What about his name, will his name be published?

**Omid:** It will be published in Iran with Behrouz's name. Again, if you try to understand this rationally it will fall apart. This is another part of the surreal nature of this whole project. They take out the Kurdish political sections, but they leave the title, which is explicitly Kurdish and explicitly political. They also take out anything referring to sex and similar sensitive topics due to the moral restrictions in Iran.

**Anna:** And the character Maysam The Whore, is that character still in there?

**Omid:** They have changed his name.

**Jurgen:** Is the fact that it will be published under his name also the reason why Behrouz accepted it? Because that would be a very difficult issue for him.

**Omid:** Behrouz originally did not want to publish it but apparently he received a lot of requests from people all over the world saying they want to read the original Persian/Farsi. But, you see, the translation was done together, and I call him one of the translators because we were working closely through the whole process. We made all the changes to the English translation, but we did not modify the original. So when he decided to put the original Persian/Farsi out into the world he had to work with an editor from the publisher to try and locate all the changes in the English and apply them to the Persian/Farsi—I do not think they identified everything because there were so many significant changes during the translation process (the Persian/Farsi version is significantly shorter than the English; writing in Persian/Farsi, Tehran-based translator and writer Araz Barseghian has published on this and many other issues, including a detailed comparative analysis of selected sections from the original alongside the English translation).[4] He even made his own changes that were not in the English. Again, this is another complicated process to describe. I do not know much about it; I was not involved. But we have a long-term plan. This Persian/Farsi publication is for now, I think he wants to make a statement and leverage for political and cultural reasons. In the future, watch this space. The uncensored version might come out (the Persian/Farsi version was rumoured to have been banned after three months in Iran and taken off the shelves in stores. However, according to press reports in Iran the fifteenth reprint was distributed in 2020 by the publisher [with the sixteenth expected soon after]; the audio book [read by award-winning actor Navid Mohammadzadeh] and the e-book are still available online—even hard copies [authorised editions by the publisher and pirated copies] are still available in many bookstores and other places. Pirated copies of the English translation are also sold in Iran).

**Onno:** Did Irwan Droog, the Dutch translator, collaborate with you at all? Were you in touch?

**Omid:** No, but I was in touch with Jurgen.

**Jurgen:** Omid did, of course, offer to help. But we did use a very good editor—two in fact. The main one is brilliant in English. One of the people who is mentioned in Omid's essay at the end of the book is a Kurdish Iraqi intellectual, Mariwan Kanie, he lives in Amsterdam and is one of my best friends. So I thought I can call him, I can cycle to meet him only two minutes away in Amsterdam-West. It sounds strange, but we did not need Omid's help translating it from English to Dutch. First, the material was very clear, the English is very clear in the book.

**Omid:** Most translators have not contacted me. The Italian, yes, we are in contact.

**Onno:** For instance, one footnote says ID which is Irwan Droog. So that is very specific for the Dutch translation, which is about that border-industrial complex. And I am not sure that every English reader would know what the BIC would refer to. But, anyway, there is definitely a different take on certain aspects of the translation.

**Omid:** But also, this book encourages people to go out and do more research and find out more about it. So these are all suggestions, sort of directions for readers.

**Supplementary Materials:** The following supporting information can be downloaded at https://www.mdpi.com/article/10.3390/h12010008/s1, Selection of Boochani's and Tofighian's Writings (in Addition to Some Collaborators).

**Funding:** This research received no external funding.

**Institutional Review Board Statement:** Not applicable.

**Informed Consent Statement:** Not applicable.

**Data Availability Statement:** Not applicable.

**Conflicts of Interest:** The authors declare no conflict of interest.

## Notes

[1] Dutch title: *Alleen de Bergen Zijn Mijn Vrienden: Verslag Vanuit de Manus-gevangenis*, published by Uitgeverij Jurgen Maas in the Netherlands on 6 November 2019.

[2] Kurdish translation by Hashem Ahmadzadeh published in 2019 by Arzan (Sweden); includes new introduction by Ahmadzadeh. In 2019 his Kurdish translation was also published by Ghazalnus in Sulaymaniyah (Slemani) in the Kurdish Region of Iraq. Ahmadzadeh has also published a scholarly article in Kurdish on *No Friend but the Mountains* and his translation.

[3] Boochani's original manuscript written in Persian/Farsi was published in Iran in 2020.

[4] Barseghian's first critique of *No Friend but the Mountains* was published soon after the Persian/Farsi version was released, it is featured in a Persian/Farsi language site: metropolatleast.ir.

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
