# Peer review of "Translation in Digital Times: Omid Tofighian on Translating the Manus Prison Narratives"

_humanities, doi:10.3390/h12010008_

Round 1

Reviewer 1 Report

This is a dialogue for the commentary section of the journal. 

the article can be improved by including a section on the Dutch translation & its reception, etc.

Author Response

Thank you for your review. I have included some extra information about the Dutch translation such as Dutch title, publication date and more.

Reviewer 2 Report

This submission consists of a transcribed interview / dialogue, and as such it is not appropriate to evaluate it according to the quality of its argumentation or structure, not could it, as a record of a conversation, be revised on these bases. The discussion is compelling, integrating Tofighian's insights on his translation of No Friend But the Mountains by Behrouz Boochani. The insights range from discussion of the economic and political contexts of the Manus prison site and larger prison industrial complex, to a fascinating set of reflections on linguistic challenges and Tofighian's solutions. The reflections on the hybrid form / genre of Boochani's book are invaluable.

On the whole, this dialogue is accessible and engaging, and likely to be of much interest to scholars of refugee-responsive literary studies, critical migration studies, as well as translation studies.

A minor revision is recommended: the film director Arash Kamali Sarvestani (director of Chauka, Please Tell Us the Time) was present for this dialogue but his voice only appears in one sentence, expressing broad agreement with a point. The very minor nature of his contribution gives the piece an unbalanced feel and it would probably be preferably simply to state that Sarvestani was present but not actively participating in the conversation.

Author Response

Thank you for your review and helpful suggestions. I have made all the necessary edits and included additional references.

Reviewer 3 Report

This is a compelling interview.  If it is the kind of work that Humanities publishes, I think that it should be published.

Author Response

Thank you for your review. I have made some necessary edits and included additional references. 

Reviewer 4 Report

This was an engaging read, touching on many of the complex questions that arise from translation and activism - and translation as activism. Passages such as the insightful (though careful) discussion about the role of human rights organisations (pp4-5) and the complicity of citizens and public institutions indicated a nuanced understanding of the  border-industrial complex, and the reflections on anti-genre and horror surrealism were helpful in framing the discussion. Overall this was an interesting and important discussion, and it's great to have a transcript of this conversation recorded in print. 

Some very minor typographical matters:

- double check grammar and phrasing for this line on p.3: "So would you be prepared to talk a little bit about 150 how your practice of translation is also intellectual collaboration"

- missing question mark: "and you were using those principles to inform your decision making."

Author Response

(The authors gave the same response as above.)
